# [Reproducibility Report] Path Planning using Neural A* Search

1 ## Reproducibility Summary

2 The following paper is a reproducibility report for "Path Planning using Neural A* Search" [1] published in ICML
3 2021 as part of the ML Reproducibility Challenge 2021. The source code [1] for our reimplementation and additional
4 experiments performed is available for running.

5 **Scope of Reproducibility**

6 The original paper proposes the Neural A* planner, and claims it achieves an optimal balance between the reduction of
7 node expansions and path accuracy. We verify this claim by reimplementing the model in a different framework and
8 reproduce the data published in the original paper. We have also provided a code-flow diagram to aid comprehension of
9 the code structure. As extensions to the original paper, we explore the effects of (1) generalizing the model by training
10 it on a shuffled dataset, (2) introducing dropout (3) implementing empirically chosen hyperparameters as trainable
11 parameters in the model, (4) altering the network model to Generative Adversarial Networks (GANs) to introduce
12 stochasticity, (5) modifying the encoder from Unet to Unet++ (6) incorporating cost maps obtained from the Neural A*
13 module in other variations of A* search.

14 **Methodology**

15 We reimplemented the publicly available source code provided by the authors in Pytorch Lightning to encourage
16 reproducibilty of the code and flexibility over different hardware setups. We reproduced the results published by
17 the authors and also conducted additional experiments. The training code was run on Kaggle with GPU (Tesla
18 P100-PCIE-16GB) and CPU (13GB RAM + 2-core of Intel Xeon).

19 **Results**

20 The claims of the original paper were successfully reproduced and validated within 3.2% of the reported values. Results
21 for additional experiments mentioned above have also been included in the report.

22 **What was easy**

23 The code provided in the original repository was well structured and documented making it easy to understand and
24 reimplement. The authors also provide the source code for dataset generation which made the task of reproducing the
25 results fairly simple .

26 **What was difficult**

27 Experimentation on some datasets took a considerable amount of time, limiting our experiments to the MP Dataset.
28 Results of runtime calculation could not be reproduced as they are affected by various factors, including dissimilarity in
29 datasets, hardware environment and A* search implementation.

30 **Communication with original authors**

31 The authors were contacted via email regarding the computational requirements of training and errors faced, to which
32 prompt and helpful replies were received.

---

[1] https://anonymous.4open.science/r/NeuralAstar-ported-6EB0/README.md

# 1 Introduction

As described in the original paper, A* search is a path planning algorithm that is frequently used in vehicle navigation, and robot arm manipulation to solve the lowest cost path problems. The paper then goes on to introduce Neural A*, a unique data-driven path planning system. The Neural A* planner uses a differentiable A* module with a fully convolutional encoder. The encoder produces a costmap, which is used by the search algorithm. The network learns to recognize visual cues from input maps which are effective in producing ground truth paths.

In this work, we reimplement the Neural A* planner (originally in Pytorch) in Pytorch Lightning [2] to make it easier to reproduce the research in the future. This allowed us to train the code flexibly over different platforms and automate the optimization process. We utilized the functionalities of PyTorch Lightning for efficient checkpointing and logging on Wandb. We check the reproducibility of the claims published in the original paper and perform multiple ablations. We then go on to explore the performance of different variations of A* coupled with the Neural A* costmap.

One of the limitations of the original work was the lack of diversity in generated paths, even though there are often multiple optimal paths. Hence in an attempt to extend the planner to account for stochastic sampling of paths we experimented with GAN [3]. We also added an implementation with Unet++ encoder [4].

# 2 Scope of Reproducibility

The original paper attempts to compare the performance of the proposed Neural A* planner with other data-driven and imitation based planners. In this work, we check the reproducibility of the following claims.

- **Comparison with other planners:** Neural A* successfully finds near-optimal paths while significantly reducing the number of nodes explored compared to other planners. The differentiable A* reformulation of the Neural A* planner is key in achieving optimal performance.
- **Start and goal maps:** The provision of start and goal maps to the Neural A* encoder enables the extraction of visual cues properly conditioned to start and goal locations, leading to significant improvement in path optimality and reduction of nodes explored.
- **VGG-16 backbone:** Using VGG-16 backbone [5] leads to better performance than ResNet-18 [6].
- **Path length optimality:** Neural A* produces nearly optimal path lengths over several datasets.
- **Planning on raw images:** Neural A* facilitates path planning on raw images without semantic pixel-wise labelling of the environment. The model outperforms several other planners in predicting pedestrian trajectories from raw surveillance data.

We validate the above claims by reproducing all results from the main article and appendix (excluding SDD dataset) except for runtime calculations (further explained under Section 6.2). In addition, we had performed several experiments to study the efficiency of the planner when coupled with different network models, mixed and shuffled datasets and variations of the simple A* search algorithm. We also performed hyperparameter tuning and implemented empirically chosen constants to instead be trainable parameters learnt through backpropagation. Section 5 contains more details.

# 3 Methodology

The Neural A* and Neural BF planners were reimplemented by us, taking inspiration from the official codebase [2]

## 3.1 Model descriptions

The simple A* search operates by maintaining an open list of nodes. The search then alternates between searching the open list for the node most likely to lead to a low cost path and expanding the list by appending the neighbours of the selected node until the goal is reached. The search uses a combination of the total best path cost and a heuristic function for node selection. The A* search algorithm has been explained in great detail in the original paper. An encoder in the Neural A* model converts a problem instance to a guidance map which assigns a guidance cost to each node. The

---

[2]https://github.com/omron-sinicx/neural-astar

Differentiable A* module then performs an A* search using the guidance cost along with the heuristic as part of the total cost. As inspired by previous chapters of MLRC [7] the codeflow for the model has been provided (Fig. 1).

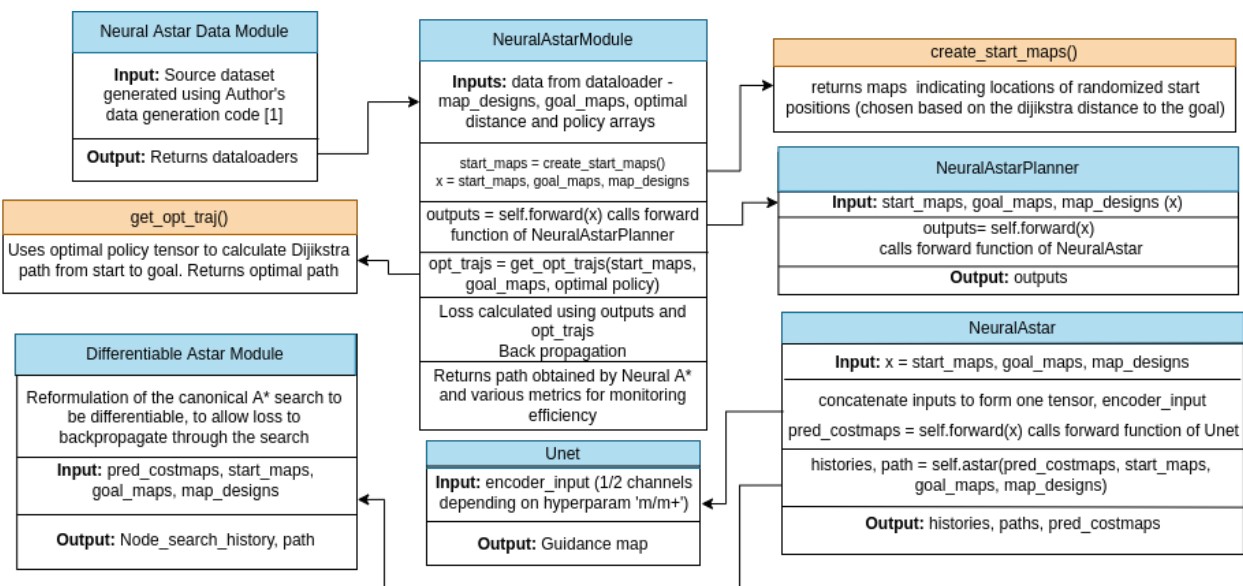

Figure 1: Codeflow of Neural A*

## Description of the planner

The Neural A* planner consists of 2 modules:

1. **Encoder :** UNet [8], a fully convolutional encoder network, is employed. Visual cues such as the contours of dead ends and by-passes are learned by the encoder. The encoder input is a concatenation of the environmental map with the start and goal maps (binary maps indicating the locations of start and goal nodes). The encoder then generates a guidance map, which is a scalar valued map representation of the problem instance. During A* search, the guidance map is used as part of the heuristic.

2. **Differentiable A* module :** The differentiable A* module is a differentiable reformulation of the classic A* search method. This was accomplished by using a discretized activation technique inspired by [9] with simple matrix operations. The module performs an A* search in the forward pass and then backpropagates the losses to other trainable modules after each epoch. This module's goal is to teach the encoder how to make guidance maps that reduce the difference between search histories and ground-truth paths.

### 3.2 Loss function

Let us consider the closed list $C$, which is a binary matrix containing all the searched nodes $V$. The optimal path P is a binary matrix containing the nodes which belong to the ground-truth path. The mean L1 loss between $C$ and the optimal path map $P$ is calculated as follows [1]:

$$L = \frac{\|C - \bar{P}\|_1}{|V|}$$

The loss defined as above penalizes both paths that stray from the optimal path (calculated by running a simple A* search, referred to as Vanilla A*) and the exploration of nodes not part of the optimal path. This loss motivates Neural A* to choose paths that are similar to the ground-truth path while also reducing node explorations.

### 3.3 Datasets

The MP, TiledMP and CSM datasets are used and preprocessed identical to the original implementation. We were unable to setup the pipeline for reimplementing results for SDD dataset due to resource constraints which is further

discussed in section 6.2. The source code[3] for generating the datasets was made publicly available by the authors. In addition, we created a mixed dataset, mixing maps from various subsets of the MP dataset to check the generalizability of the model. This is described further in section 5.1.

## 3.4 Hyperparameters

The hyperparameters and their respective ranges are same as mentioned in the original paper. The hyperparameters were set after conducting random search using Wandb [10]. The best hyperparameters are *g_ratio* = 0.2 and rest of the hyperparameters same as the paper. As can be observed from the values, the changes in hyperparameters are in accordance with weighted A*, decreasing the weight of the heuristic, the number of explorations and path optimality increase. On the other hand increasing the weight of the heuristic leads to a decrease in both number of explorations and path optimality. Encoder input can be of the type **m+** (includes start maps, goal maps and map designs), or **m** (includes just the map designs). It can be seen that Exp and Opt (Section 3.5) have an inverse relationship, and hence it is important to find hyperparameters that maintain the equilibrium between them. Note that the original paper does not report results for Resnet backbone with Encoder input of type *m*.

Table 1: Results with the Best Hyperparameters

| Input Type | Version | Opt | Exp | Hmean | Suc |
|---|---|---|---|---|---|
| m | Best | 64.95 (60.22, 69.87) | **42.80(39.52, 46.07)** | **45.44(42.03, 48.90)** | 100.0 |
| | Original | **67.0 (65.1, 68.8)** | 36.8(35.6, 38.1) | 41.5(40.2, 42.7) | 100.0 |
| m+ | Best | 78.97(75.23, 82.97) | **44.49 (41.28, 47.68)** | **52.42 (49.14, 55.77)** | 100.0 |
| | Original | **87.7 (86.6, 88.9)** | 40.1(38.9, 41.3) | 52.0(50.7, 53.1) | 100.0 |

Table 2: Resnet Backbone with best hyperparameters

| Input Type | Version | Opt | Exp | Hmean | Suc |
|---|---|---|---|---|---|
| m | Best | 64.10 (59.34, 69.07) | 42.40 (39.14, 45.76) | 44.60 (41.15, 48.12) | 100.0 (100.0, 100.0) |
| | Original | — | — | — | — |
| m+ | Best | 75.57 (71.88, 79.83) | **43.71 (40.57, 46.85)** | **50.80 (47.77, 53.93)** | 100.0 (100.0, 100.0) |
| | Original | **79.8(78.1, 81.5)** | 41.4(40.2, 42.7) | 49.2(47.9, 50.5) | 100.0 (100.0, 100.0) |

## 3.5 Experimental setup and code

The models were trained using the RMSProp optimizer with a batch size of 100 with a learning rate 0.001. The models were trained for 100 epochs on the MP Dataset and 400 epochs for the CSM and TiledMP datasets [11; 12; 13]. The following metrics are as defined in the original paper and evaluated for each trained model on their test set performance.

- **Path optimality ratio (Opt)**: measures the percentage of shortest path predictions for each map.

- **Reduction ratio of node explorations (Exp)**: measures the number of search steps reduced by a model as compared to vanilla A* search. Let $E^*$ be the number of nodes explored by vanilla A* and $E$ be the number of nodes explored by a particular model. Then Exp is defined as $\max(100(\frac{E^*-E}{E^*}, 0))$.

- **Harmonic mean of Opt and Exp (Hmean)**: Indicates the trade-off between Opt and Exp achieved.

A higher Hmean value is desired and indicates that a model achieves satisfying optimality and search efficiency. The central planners implemented in this work are Neural A* and Neural BF (i.e. Neural Best First which is simply the Neural A* planner with hyperparameter g_ratio set to 0). Various other planners were used as baselines for judging the performance of Neural A* including SAIL [14], BBA* [15; 1], Weighted A* [16], Best First Search, VIN [17], GPPN [18] and MMP [19]. The values of these metrics for baseline planners were obtained from running the official source code and were identical to those published in the original paper.

---

[3]https://github.com/omron-sinicx/planning-datasets

## 3.6 Computational requirements

The training code was run on Kaggle with GPU (Tesla P100-PCIE-16GB) and CPU (13gb RAM + 2-core of Intel Xeon). The average training times were 1.25, 72 and 58 hours for the MP, Tiled MP and CSM datasets respectively.

# 4 Results

Detailed description of the experiments and results supporting the claims made in Section 2 is given below

## 4.1 Results reproducing original paper

We compare the Bootstrap means and 95% confidence intervals of path optimality ratio, reduction ratio of node explorations and the harmonic mean of these values from the ported code (**R**) and original paper (**O**).

Table 3: Ablation Results

| Dataset | Planner | Version | Opt | Exp | Hmean | Suc |
|---------|---------|---------|-----|-----|-------|-----|
| MP | Neural A* | R | 86.15 (85.00, 87.37) | 39.96 (38.74, 41.18) | 51.50 (50.23, 52.80) | 100.0 |
| | | O | **87.7 (86.6, 88.9)** | **40.1 (38.9, 41.3)** | **52.0 (50.7, 53.3)** | 100.0 |
| | Neural BF | R | 74.29 (72.70, 75.90) | 44.97 (43.66, 46.27) | 50.93 (49.64, 52.23) | 100.0 |
| | | O | **75.5 (73.8, 77.1)** | **45.9 (44.6, 47.2)** | **52.0 (50.7, 53.4)** | 100.0 |
| Tiled MP | Neural A* | R | **74.84 (71.20, 78.69)** | 48.86 (45.25, 52.44) | **55.66 (52.56, 58.73)** | 100.0 |
| | | O | 63.0 (60.7, 65.2) | **55.8 (54.1, 57.5)** | 54.2 (52.6, 55.8) | 100.0 |
| | Neural BF | R | **44.17 (40.95, 48.34)** | 61.3 (57.74, 64.92) | **44.61 (39.37, 49.00)** | 100.0 |
| | | O | 43.7 (41.4, 46.1) | **61.5 (59.7, 63.3)** | 44.4 (42.5, 46.2) | 100.0 |
| CSM | Neural A* | R | **77.82 (74.07, 81.75)** | 34.99 (31.06, 38.86) | 42.64 (38.79, 46.51) | 100.0 |
| | | O | 73.5 (71.5, 75.5) | **37.6 (35.5, 39.7)** | **43.6 (41.7, 45.5)** | 100.0 |
| | Neural BF | R | **82.70(81.29, 84.15)** | 39.16 (37.92, 40.38) | **49.27 (47.98, 50.55)** | 100.0 |
| | | O | 79.8 (78.1, 81.5) | **41.4(40.2, 42.7)** | 49.2(47.9, 50.5) | 100.0 |

### 4.1.1 Path length optimality

Path length optimality is another metric used to judge the performance of the planner. Path length optimality P is defined as the percent ratio of optimal path length and predicted path length.

Table 4: Path length optimality comparison

| Dataset | Planner | Reimplementation | Original Paper |
|---------|---------|------------------|----------------|
| MPD | Neural A* | **99.23 (98.99, 99.51)** | 99.1 (99.0, 99.2) |
| | Neural BF | **98.04 (97.65, 98.45)** | 97.5 (97.3, 97.8) |
| TiledMP | Neural A* | **99.12 (98.89, 99.39)** | 98.4 (98.3, 98.6) |
| | Neural BF | **96.85 (96.25, 97.47)** | 95.0 (94.7, 95.4) |
| CSM | Neural A* | **99.31 (99.09, 99.55)** | 98.9 (98.8, 99.0) |
| | Neural BF | **97.62 (97.14, 98.13)** | 97.4 (97.1, 97.6) |

### 4.1.2 Ablation Results from the Original Paper

Comparision of results of reproducing ablations from original paper.

1. The encoder was not provided start and goal node locations
2. Resnet-18 backbone was used instead of VGG-16

Table 5: Ablation results comparison

| Ablation | Version | Opt | Exp | Hmean | Suc |
|----------|---------|-----|-----|-------|-----|
| Ablation 1 | R | **70.07 (68.14, 72.01)** | **39.56(38.27, 40.83)** | **45.89(44.54, 47.27)** | 100.0 |
| | O | 67.0 (65.1, 68.8) | 36.8(35.6, 38.1) | 41.5(40.2, 42.7) | 100.0 |
| Ablation 2 | R | **82.70(81.29, 84.15)** | 39.16 (37.92, 40.38) | **49.27 (47.98, 50.55)** | 100.0 |
| | O | 79.8 (78.1, 81.5) | **41.4(40.2, 42.7)** | 49.2(47.9, 50.5) | 100.0 |

# 5  Additional Experiments

For additional experiments we suggest the use of "number of explorations" as an additional metric. Since Exp is a relative metric that compares expansions by a particular model with Vanilla A*, the degraded performance of Vanilla A* can sometimes cause misleading results. On the other hand, "number of explorations" is an intuitive and independent metric making it easier to compare models. The author also suggested the use of number of explorations as a proxy metric that is likely to be used for neural planners. We use NA* Exps as number of explorations by Neural A* and VA* Exps as number of explorations by Vanilla A*.

## 5.1  Shuffling Dataset

In the original implementation, the authors train the Neural A* model on subsets of the MP dataset as individual environments. This leads to the model working efficiently on few maps, but generalisation performance of the model suffers. In this experiment we choose some maps from each of the MP subsets and create a mixed set. We then train the model on this dataset and compare it's results with those on individual environments. From Table 6 we observe that the model achieved better performance on a generalized test set as compared to individually trained models.

Table 6: Mixed set comparision

| Sub-dataset | NA* Exps | VA* Exps | Opt | Exp | Hmean |
|---|---|---|---|---|---|
| alternating gaps | 58.51 | 102.89 | 62.24 (56.06, 68.85) | **36.21 (32.48, 39.83)** | 38.14 (33.96, 42.34) |
| bugtrap forest | 76.23 | 98.73 | 73.14 (68.51, 78.13) | 24.93 (21.52, 28.20) | 32.38 (28.82, 35.99) |
| forest | 66.56 | 91.15 | 73.80 (69.62, 78.22) | 27.18 (24.13, 30.09) | 35.69 (32.22, 39.11) |
| gaps and forest | 61.77 | 89.80 | 71.48 (66.95, 76.16) | 30.35 (26.84, 33.92) | 39.20 (35.42, 42.92) |
| mazes | 79.79 | 104.95 | 71.83 (67.19, 76.68) | 27.10 (23.28, 30.72) | 34.84 (30.92, 38.82) |
| multiple bugtraps | 75.04 | 99.11 | 77.39 (73.20, 81.89) | 25.70 (22.10, 29.17) | 33.44 (29.61, 37.30) |
| shifting gaps | 61.04 | 99.55 | 70.56 (65.17, 76.27) | 32.56 (28.52, 36.78) | 38.70 (33.96, 43.39) |
| single bugtrap | 101.8 | 94.02 | 71.91 (66.95, 77.04) | 18.11 (14.60, 21.27) | 23.13 (19.44, 26.82) |
| Average for MP dataset | 72.59 | 97.52 | 71.44 (69.65, 73.30) | 27.29 (26.00, 28.60) | 33.98 (32.53, 35.40) |
| Mixed set | 57.74 | 94.06 | **84.32 (80.50, 88.39)** | 35.51 (32.05, 38.85) | **45.97 (42.25, 49.71)** |

## 5.2  Dropout

Experimented by adding dropout to UNet taking inspiration from [20] where it is claimed that the dropout introduces stochasticity at the planning step. We believe that further performance improvements can be made by implementing a replanning step as explained in section 7.

Table 7: Dropout Results for g_ratio = 0.5

| Input Type | Version | NA* Exps | VA* Exps | Opt | Exp | Hmean | Suc |
|---|---|---|---|---|---|---|---|
| m | R | 66.63 | 103.01 | **72.58 (68.25, 77.17)** | 31.43(28.14, 34.65) | 38.80(35.34, 42.23) | 100.0 |
|  | O | 68.18 | 103.01 | 67.00 (65.10, 68.80) | **36.80(35.60, 38.10)** | **41.50(40.20, 42.70)** | 100.0 |
| m+ | R | 59.08 | 103.01 | 84.71(81.61, 88.05) | 37.89(34.83, 40.92) | 49.03(45.89, 52.21) | 100.0 |
|  | O | 56.27 | 103.01 | **87.70 (86.60, 88.90)** | **40.10(38.90, 41.30)** | **52.00(50.70, 53.10)** | 100.0 |

## 5.3  Suggested Experiments by the Authors

The authors have mentioned in the paper that if there are several paths to the goal location, the planner fails to identify actual pedestrian trajectories. They also suggested that adopting a generative framework [21] that permits numerous paths to be stochastically sampled could be a suitable extension to address this issue [22]. While communicating with the authors, we were suggested to experiment by changing the encoder to a generative model based architecture like GAN. The authors also suggested that learning the value of temperature via backpropagation is an interesting direction to explore.

### 5.3.1  Temperature Function

The temperature function $\tau$ used in the differentiable A* module is empirically chosen by the authors. In this experiment we implement $\tau$ as a trainable parameter of the Neural A* model whose value gets updated by backpropagation. It

was observed that on initialization of $\tau$ to the square root of map size, the value remains almost constant over the training process. The sweeps have only been conducted on one sub dataset of the MP dataset (bugtrap_forest_32) due to computational restraints.

Table 8: Hyperparameter Sweep on Temperature Function with g_ratio = 0.2

| Temperature | NA* Exps | VA* Exps | Opt | Exp | Hmean | Suc |
|---|---|---|---|---|---|---|
| $\sqrt{32}$(Original) | 46.02 | 69.63 | 78.32 ( 75.02, 81.84) | 38.95(36.37, 41.32) | 50.15(47.71, 52.63) | 100.0 |
| $2\sqrt{32}$ | 46.48 | 84.22 | 78.20 ( 74.30, 82.31) | 36.50(33.50, 39.39) | 46.64 (43.73, 49.52) | 100.0 |
| $3\sqrt{32}$ | 52.96 | 75.84 | 73.13(68.95, 77.60) | **41.98(39.53, 44.43)** | 50.64 (47.92, 53.34) | 100.0 |
| $4\sqrt{32}$ | 45.00 | 81.34 | 75.41(71.61, 79.44) | 41.34(38.71, 43.86) | **50.94 (48.31, 53.61)** | 100.0 |
| $\sqrt{32}/2$ | 57.51 | 88.85 | 79.50 (75.74, 83.53) | 31.87 (29.48,34.18) | 43.20(40.60, 45.80) | 100.0 |
| $\sqrt{32}/3$ | 50.05 | 76.61 | **85.33 (82.21, 88.86)** | 32.46 (30.22, 34.67) | 45.10(42.46, 47.76) | 100.0 |

### 5.3.2 GANs

General Adversarial Networks are a class of generative models that involves two sub-models, a Generator which generates new examples from the domain and a Discriminator which classifies the generated output as real or generated. Generative models have been used for path planning before [23]. For our experiments we use the vanilla GANs from Pytorch Lighting-Bolts [24] We believe that in cases with multiple possible routes the inherent stochasticity of the generator would help in producing guidance maps that tempt the planner to try different possibilities and make it more robust. From the results mentioned in Table 9 and 10 we can infer that our GANs experiment combined with best hyperparameters (g_ratio = 0.5) outperforms the original paper for maps with input type as *m*. Further, for the same hyperparameters, it can be seen that GANs with input type as *m* performs better than that with *m+*, contrary to the original paper.

## 5.4 UNet++[4]

As advised by the author, we experimented with UNet++ architecture while undertaking an architectural search and assessing its scope for further performance improvements. UNet++ is similar to UNet, with an ensemble of variable depth UNets which partially share an encoder and co-learn simultaneously using deep supervision. However, we find that UNet performs better than UNet++, showing that powerful segmentation models might not always be better at learning guidance maps for Neural A*.

Table 9: Encoder Architecture Experimentation on MP Dataset with 'm' type encoder input

| Encoder Type | NA* Exps | VA* Exps | Opt | Exp | Hmean | Suc |
|---|---|---|---|---|---|---|
| UNet | 68.18 | 103.01 | 67.00 (65.10, 68.80) | 36.80(35.60, 38.10) | 41.50(40.20, 42.70) | 100.0 |
| UNet++ | 64.99 | 102.75 | **76.41(72.27, 80.79)** | 31.79(28.52, 35.00) | 39.68(36.38, 43.02) | 100.0 |
| GAN | 68.40 | 103.25 | 66.44(61.53, 71.54) | **38.27 (34.81, 41.68)** | **42.12(38.6, 45.65)** | 100.0 |

Table 10: Encoder Architecture Experimentation on MP Dataset with 'm+' type encoder input

| Encoder Type | NA* Exps | VA* Exps | Opt | Exp | Hmean | Suc |
|---|---|---|---|---|---|---|
| UNet | 56.27 | 103.01 | **87.70 (86.60, 88.90)** | **40.10(38.90, 41.30)** | **52.00(50.70, 53.10)** | 100.0 |
| UNet++ | 57.13 | 102.75 | 87.46(84.67, 90.48) | 36.41(33.30, 39.49) | 48.17(44.97, 51.46) | 100.0 |
| GAN | 67.74 | 103.25 | 65.54(60.60, 70.68) | 38.44(34.97, 41.86) | 41.74(38.20, 45.31) | 100.0 |

## 5.5 Variations of A*

In this experiment we test the performance of various versions of A* search coupled with and without the costmaps generated by the Neural A* module. We can infer from the obtained results that the costmaps generated by the Neural A* planner can be used with other variations of the A* search. However, a particular model works best with variations of A* using the weight of heuristic similar to the model. As a result, we decided against incorporating the costmaps generated using dynamic weighted A* because the model's weight varies while the search is ongoing.

Table 11: Variations of A* search on **mazes** (MP Dataset)

| A* Variation | Opt | Exp | Hmean |
|---|---|---|---|
| Vanilla A*[25] | 57.99 (54.06, 62.01) | **21.48 (18.86, 23.95)** | 24.71 (21.02, 28.29) |
| Weighted A* | 44.45 (39.56, 49.36) | 17.80 (13.38, 22.04) | 18.15 (14.92, 21.18) |
| Beam search[26] | **58.74 (55.04, 62.43)** | 20.38 (17.85, 22.78) | **26.07 (22.69, 29.40)** |

# 6  Discussion

Our results reproduce the findings of [1] and strengthen the claims made in section 2. From section 4.1 we conclude that the Neural A* planner significantly reduces the number of nodes explored while still maintaining sufficiently large path optimality. The results obtained from the reimplementation were within 3.2% of the official implementation. From section 4.1.2 we also validate that provision of start and goal position to the encoder plays a crucial role in guidance map generation. We also conclude that ResNet-18 architecture leads to a drop in performance. For additional experiments (Section 5), we have worked on the suggestions of the authors. The generalizability of the model was displayed by the use of generic datasets and coupling the planner with different A* implementations. We add UNet++ and GANs implementations and experiment with dropout.

## 6.1  What was easy

Porting the code to Lightning was easy as the original source code is very structured and well commented. Ablations reported in the paper are well organized, and results are clearly reported making them easier to replicate. Writing the code for additional experiments was also not difficult as the original code is written well, allowing for fast execution.

## 6.2  What was difficult

Runtime calculations posed many difficulties due to differences in training datasets, hardware and A* implementation. The paper had several parameters that could be tuned hence it was difficult to ascertain their relative importance. Experimentation was not possible on SDD Dataset which composed of raw images due to training time constraints. Similarly, the TiledMP and CSM datasets also took considerable time to train, and hence we only reproduced the experiments from the original paper on these sets. All other experiments were conducted on the MP dataset.

## 6.3  Communication with original authors

The conversation was based on the difficulties mentioned above along with advice on how to approach the future improvements mentioned in the paper. The authors' replies were very prompt and thorough, enabling us to better reproduce their work. We also implemented several of their suggestions, as explained in Section 5.3.

# 7  Future Work

For future reproduction of this paper we suggest the use of "number of explorations" as an additional metric. Exp and Hmean values alone can often be misleading and difficult to analyze. On the other hand, the number of explorations is an intuitive metric and makes it easier to compare models. The author also suggested the use of number of explorations as a proxy metric that is likely to be used for neural planners. We list down few experiments which were planned but couldn't be completed. We believe that future works can be built on these two extensions:

- **Stochastic Path Sampling in Neural A* :** For this, we believe that implementing a replanning step coupled with dropout as suggested in [20] will be an interesting addition to the planner. Similar to GANs, Variational Auto Encoders can also be used. This would force the planner to propose multiple paths for the same start and goal positions, thus overcoming its current limitation.

- **Extension of Neural A* to high dimensions:** For high dimensional spaces we think that using an architecture like 3D UNet [27] or adding a feature extractor similar to [28] can be a plausible solution.

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
