# OpenReview forum: "[Reproducibility Report] Path Planning using Neural A* Search"
_ML_Reproducibility_Challenge/2021/Fall — Reject_

### Official Review · Reviewer_ovGD · 2022-02-16
**An apparently thorough reproduction effort, but hampered by presentation issues**

**Rating:** 5
**Confidence:** 4

**Review:**

This reports details efforts in reproducing the paper "Path Planning using Neural A* Search", namely:
- A reimplementation of the original codebase into PyTorch Lightning, to benefit from its standardization and tooling features
- A reproduction of the paper's main results and ablations
- Investigations of several variants, originating from discussions with the original paper's authors

This is overall a solid reproduction effort: even if, as acknowledged in this report, the original codebase was clean and easy to use (making the port to Lightning relatively straightforward), the empirical evaluation is thorough and several non-trivial variants have been implemented and evaluated (going beyond simple hyperparameter sweeps).

Unfortunately, I found the presentation to be lacking, making several parts of the report difficult to understand, without clear takeaways. This is why I am currently leaning towards rejection. At high level, this report isn't sufficiently self-contained: I assume several points would be clearer after reading the original paper, but the report should introduce the relevant concepts and set proper context. Here are some specific examples:
1. The codeflow in Fig.1 is quite hard to follow (notations should be more consistent and one should be able to more easily track execution line by line)
2. What is \bar{P} in the definition of L in Section 3.2?
3. Several quantities are used before they are defined (or just not defined at all), for instance: g_ratio / Exp / Opt / Hmean / number of explorations
4. The following sentence is hard for me to understand: "As can be observed from the values, the changes in hyperparameters are im accordance with weighted A*, decreasing the weight of the heuristic, the number of explorations and path optimality increase"
5. In Table 1, "Best" refers to the best what? (note that the Table's title already says "Best Hyperparameters")
6. Table 2 is with Resnet Backbone => I assume Table 1 is with VGG? (this isn't mentioned)
7. There is no explanation on what the reported metrics "X (Y, Z)" represent. I assume X is the mean or median, and Y-Z some confidence interval? Over how many runs? And how can we say that any method is better than another when so many intervals have significant overlaps? (for instance claiming that "GANs with input type as m performs better than that with m+" based on results in Table 9 & 10 seems unfounded, as far as I can tell it performs quite similarly)
8. In Section 5.1 it is not clear to me what is the objective and what conclusion we can draw. It seems unsurprising that a model trained on a larger variety of maps would generalize better, but I am not even sure the comparison makes sense here, since as far as I understand the models aren't tested on the same maps.
9. What do we learn from the dropout experiments? Are both R and O using dropout? Why use g_ratio=0.5?
10. Table 8 reports a hyper-parameter sweep over tau, but the section is about learning tau. What's going on there?
11. The GAN experiment sounds interesting but could use more details. On what data is the GAN trained? How is it used exactly in the Neural A* algorithm? Do we learn anything interesting here?
12. Section 5.5 mentions testing "various versions of A* search coupled with and without the costmaps generated by the Neural A* module". But Table 11 only has one row per A* variant, is it with or without these costmaps? Note also that the numbers in this table seem pretty low compared to the maze results from Table 6, why is that?
13. It is claimed that the reimplementation gave results "within 3.2% of the official implementation". How was this number obtained?

(side note about a minor typo found a few times: "comparision")

Overall I encourage the authors to better contextualize their experiments and highlight their findings, so that the report can be read as a meaningful extension of the original work, rather than just "we reproduced the results and launched a few more experiments along the way".

---

### Official Review · Reviewer_oPB9 · 2022-03-07
**Convincing repro and report**

**Rating:** 7
**Confidence:** 4

**Review:**

The authors reproduced Neural A*, by reimplementing it and basing/comparing it on the orignal authors (Yonetani et al.). There are significant number differences (e.g. Tables 3, 5, 7) but the authors noted that the datasets are not exactly the same. The reproduction is nevertheless convincing and validate the experimental claims of Yonetani et al. The reproducibility report itself is detailed enough and easy to follow. It would be useful for the reproduction's code and artifacts to be pushed to a public repository, if possible.

---

### Meta-Review · Area_Chair_xcE6 · 2022-04-08

**Recommendation:** Reject
**Confidence:** 5

**Metareview:**

While the paper is a decent reproducibility effort, I agree with reviewer ovGD that several aspects of presentation needs work. The empirical results although will be very beneficial for the understanding of the original paper.

---

### Decision · Program_Chairs · 2022-04-09

Reject